# Giant switchable non thermally-activated conduction in 180° domain walls in tetragonal Pb(Zr,Ti)O₃

Felix Risch [1], Yuri Tikhonov[2], Igor Lukyanchuk[2], Adrian M. Ionescu [1] & Igor Stolichnov [1] ✉

Conductive domain walls in ferroelectrics offer a promising concept of nanoelectronic circuits with 2D domain-wall channels playing roles of memristors or synoptic interconnections. However, domain wall conduction remains challenging to control and pA-range currents typically measured on individual walls are too low for single-channel devices. Charged domain walls show higher conductivity, but are generally unstable and difficult to create. Here, we show highly conductive and stable channels on ubiquitous 180° domain walls in the archetypical ferroelectric, tetragonal Pb(Zr,Ti)O₃. These electrically erasable/rewritable channels show currents of tens of nanoamperes (200 to 400 nA/μm) at voltages ≤2 V and metallic-like non thermally-activated transport properties down to 4 K, as confirmed by nanoscopic mapping. The domain structure analysis and phase-field simulations reveal complex switching dynamics, in which the extraordinary conductivity in strained Pb(Zr,Ti)O₃ films is explained by an interplay between ferroelastic a- and c-domains. This work demonstrates the potential of accessible and stable arrangements of nominally uncharged and electrically switchable domain walls for nanoelectronics.

Conductive domain walls (DWs) in ferroelectrics attract a lot of attention as potential basic elements for novel paradigms of electronics[1–4]. The ability of domain walls to form nonvolatile conductive channels, which can be moved, erased and recreated via electrical stimuli, and their plasticity hold promise for a variety of applications from memories to neuromorphic circuits. Since the first report of electrical transport properties of domain walls in BiFeO₃ (BFO) films[5] an impressive progress has been made in exploring different materials with domain wall conduction, demonstrating their functionalities[6–10] and advancing their technological relevance[11]. A nonvolatile ferroelectric domain wall memory has been proposed based on the formation of domain walls bridging two planar electrodes on a BFO film with insulating bottom interface[12]. In an alternative concept, temporary-formed domain wall channels were used for nondestructive readout of persistent polarization states[13]. Beyond BFO

systems, multi-level DW-based memristor functionalities have been demonstrated on thin slabs of LiNbO₃, an uniaxial ferroelectric with conically shaped domains[14,15]. In a number of other materials including Pb(Zr,Ti)O₃ (PZT)[16], BaTiO₃[17] and ErMnO₃[18], domain wall conduction has been reported, however its relevance for logic and memory devices remains to be validated.

The physical origin of domain wall conduction involves either intrinsic or extrinsic mechanisms. The former implies a change of the energy band structure occurring at the domain boundary due to a polarization discontinuity (charged DWs), while the latter is associated with defect accumulation such as oxygen vacancies within the domain wall region. Generally, the extrinsic mechanism is applicable to neutral domain walls such as 180°-DWs in tetragonal PZT, where the role of oxygen vacancies in the DW-transport properties was experimentally proven[19]. However, the physics of domain wall conduction becomes

[1]Nanoelectronic Devices Laboratory (NanoLab), Ecole Polytechnique Fédérale de Lausanne (EPFL), 1015 Lausanne, Switzerland. [2]Laboratory of Condensed Matter Physics, University of Picardie, 80039 Amiens, France. ✉e-mail: igor.stolichnov@epfl.ch

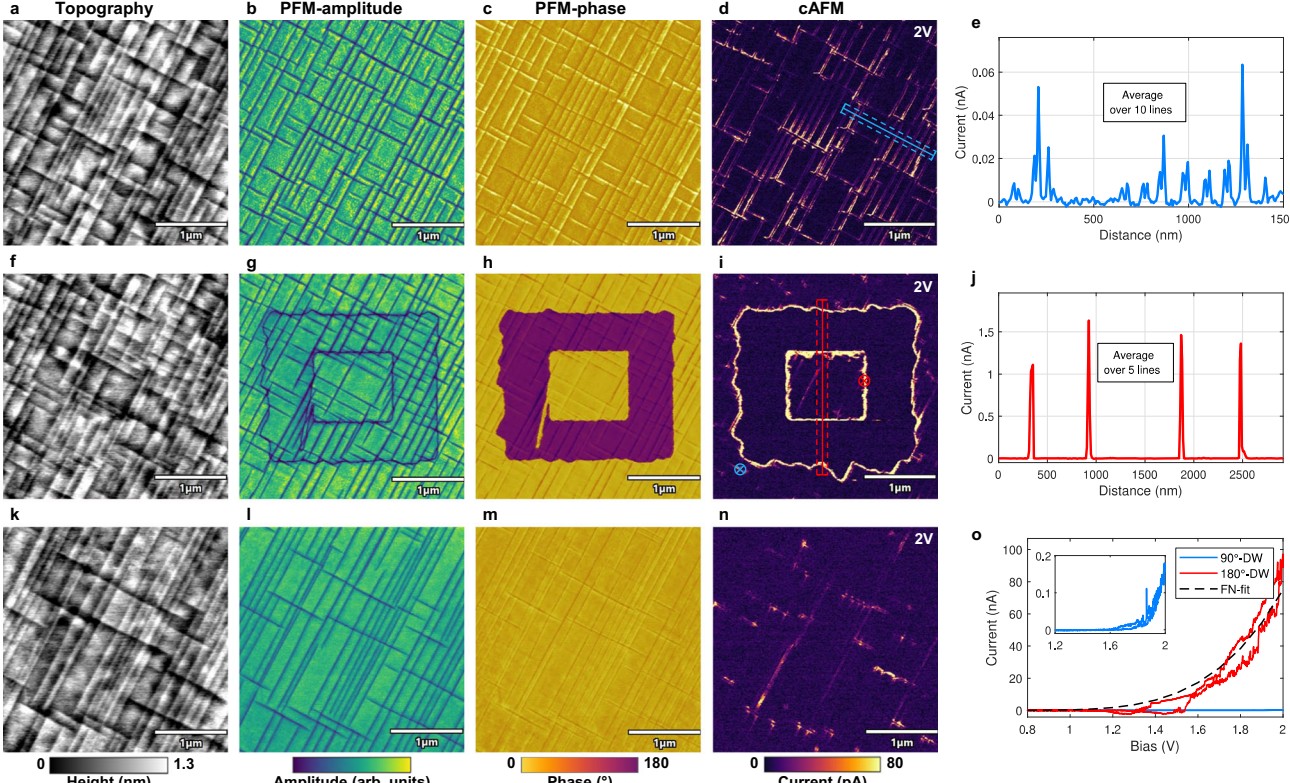

**Fig. 1 | Reversible electrical writing of 180°-DWs and their conduction response.** The three rows of images represent sequential scans of the same area (the a-domain crosshatch pattern being altered after each scan), cAFM maps are acquired with a 2V bias. **a–d** AFM-tapping mode topography, PFM and cAFM maps of a pristine area of the PZT film. The phase image confirms the uniform c-domain polarization and the cAFM maps reveals weak conduction traces at the position of 90°-DWs. **f–i** The same area after poling by an AFM-tip with −5 V tip bias applied. The switched 180° domains (purple regions in the phase map) are outlined by a high current response in the cAFM image. **k–n** Rescan of the same area after poling it back into the initial state by applying a tip bias of 5 V. The a-domain pattern visibly changes under the applied fields, the c-domains are poled back to the original state and no current at the former 180°-DW positions is detected. **e, j** Cross-section current profiles of cAFM maps as indicated in **d** and **i** averaged over 10 (blue) and 5 (red) lines respectively. **o** I–V characteristics probed by the AFM-tip that contacts the surface at the positions indicated in **i** on a 90°-DW (blue) and 180°-DW (red) together with the FN-fit (black).

more complex if strain and other factors, resulting in domain distortion, are taken into account. In particular, it was shown that nominally neutral ferroelastic 90°-DWs could become bent and hence partially charged due to an interplay between the lattice tetragonality and mismatch with the substrate. Consequently, the 90°-DWs in such PZT films exhibited an intrinsic conductivity with metallic-like non thermally-activated transport characteristics[20]. Despite the significant advancements in understanding the mechanisms of domain wall conduction and proofs of concepts of DW-based devices, important issues still impede progress in the field. A relatively low level of current sustained by individual domain walls (typically within the picoampere range) is too low for domain wall based electronics. A possible way to address this problem is using multiple domain walls[21,22], but in this case the scaling potential is compromised. Charged domain walls can provide much higher currents[23,24], however they are generally unstable and form transiently under voltage[13], or require a special poling procedure[25,26], which complicates their practical use. In Supplementary Table 1, we have summarized and compared domain wall conduction values reported in previous works together with the related DW-type, poling procedures and other material specific remarks.

Here, in the quest for robust, accessible and controllable DW-based conductors we focus on tetragonal Pb(Zr,Ti)O₃ (PZT), one of the archetypical ferroelectrics offering well-studied domain structures and mature processing technologies. We demonstrate that high-magnitude switchable non thermally-activated domain wall conduction can be achieved in nominally neutral 180°-DWs. The transport properties of the domain walls are revealed by scanning probe

characterizations, which combine atomic force microscopy (AFM) topography imaging, piezoelectric force microscopy (PFM), conductive AFM (cAFM) and cryogenic scanning probe experiments down to 4 K. The on-demand created domain walls were utilized to show basic memristive device functionalities by controlling their position through electrical stimuli. A large degree of control, good stability and high magnitude of conductance of such single domain walls hold promise for readily accessible tunable multi-resistive state devices for neuromorphic applications.

## Results

### Switchable conduction in nominally neutral 180°-DWs

60 nm PZT (Zr/Ti = 10:90) layers were epitaxially grown on a DyScO₃ (DSO) substrate with a 20 nm SrRuO₃ (SRO) bottom electrode by pulsed laser deposition (PLD) (details in "Methods"). The PZT film showed predominantly downward-oriented c-domains interrupted by thin 10 nm wide ferroelastic a-domains formed due to the specific strain conditions as described in ref. 27. As seen in the AFM topography and PFM images (Fig. 1a–c), these in-plane a-domains form a rectangular crosshatch pattern separating the uniformly polarized c-domains (details in "Methods"). cAFM maps collected with a sample bias of 2 V revealed conductive traces that follow this a-domain pattern (Fig. 1d). The detected current at the 90°-DWs reached 10–30 pA, whereas c-domains showed an insulating behavior (Fig. 1e), in agreement with previous results obtained on similar PZT samples[20]. I–V characteristics of the 90°-DWs revealed a strong asymmetric behavior with no current detectable under negative sample biases (see

Supplementary Fig. 1a). Since the polarization of the as-grown PZT film was downward-oriented, positive biases applied to the SRO bottom electrode promoted switching of the ferroelectric domains. Therefore, c-domains could be switched with a sample bias above the coercive field of ~5 V (see Supplementary Fig. 1b) as shown in Fig. 1f–i, representing the same area rescanned after poling. The cAFM scan in Fig. 1i shows remarkably strong conduction traces outlining all c-domain boundaries with nA-range currents observed in the averaged cross-section profile (Fig. 1j). The conduction was polarity-dependent, with no detectable current under a negative sample bias identical to the 90°-DWs (see Supplementary Fig. 1c). After poling, the switched c-domains slightly shrank down over time and reached their final configuration by adopting - wherever possible - the boundaries defined by the ferroelastic a-domains (Fig. 1g–i). Once they reached their stable position, the conductive 180°-DWs remained steady with no sign of conduction decrease for several days. Tens of consecutive scans under the same sample bias confirmed the positional stability of the domain walls, and eliminate the possibility of polarization switching charges (see Supplementary Fig. 2) as the underlying conduction mechanism. Furthermore, the 180°-DWs formed at room temperature (RT) persisted up to 100°C and showed non thermally-activated conduction properties, i.e. the domain wall conduction decreased with increasing temperature (see Supplementary Fig. 3). The averaged cross-sections of the conduction traces and I–V curves confirmed that the written 180°-DWs produce consistently high currents that are up to three orders of magnitude higher compared to the pristine 90°-DWs (Fig. 1e, j, o). The essential features of the conduction behavior, including its polarity dependence and non-linear I–V characteristics, are consistent with the transport limited model described by electron tunneling injection through the probe/PZT interface[28]. The Fowler-Nordheim (FN) formalism provides an adequate description of this mechanism, which is supported by the FN-fit of the I–V curve (black dotted line) in Fig. 1o (details in Supplementary Note 1).

Upon complete removal of the switched domains by backpoling to the initial state (Fig. 1k–m), the conductance vanished, without leaving any conductive traces at the former position of the 180°-DWs (Fig. 1n). This shows the degree of control over the conductive channels, which can be created, modified, erased and recreated on demand, by an external voltage.

## Domain wall based devices

To further investigate the relevance of the conductive channels for device applications, domain wall contacting and conduction measurements with a parallel-plate capacitor geometry were performed. The acquired data were consistent with the data collected on the bare PZT surface and basic device functionalities could be demonstrated based on controlling the presence and position of the domain walls by electrical stimuli. Cr/Au (5/20 nm thick) electrodes with a size of $250 \times 250$ nm$^2$ were used for contacting/controlling the individual 180°-DWs. Figure 2a–c shows PFM and cAFM scans, in which the conductive 180°-DW was pushed below an electrode, thus connecting the top and bottom electrode and bringing the device into a low-resistance state (LRS). Sequential I–V curves (Fig. 2d), performed by placing the conductive tip onto the top electrode, confirmed a maximum current of 50 nA for 2 V. To prove that the measured conduction originates purely from the DW-transport, the same measurements were repeated after driving the domain wall outside of the capacitor via a DC bias. The negative/positive sample bias caused the domain to shrink down (Fig. 2e–h) or to expand (Fig. 2i–l) beyond the electrode boundary leaving the capacitor uniformly poled down- or upwards, hence bringing the device back to its initial high-resistance state (HRS). Regardless of the polarization direction, the conduction decreased by more than three orders of magnitude compared to the LRS (Fig. 2d, h, l). Recurrent movement of the domain wall in- and outside of the capacitor resulted in a reproducible conduction-recovery/

suppression. The time-resolved measurements under repeated non-destructive readout pulses permitted further insights into the DW-device characteristics (Fig. 2m, n). Figure 2m shows the domain wall current collected from the top electrode using a train of 10 (2 V/200 ms) triangular pulses. A sequence of 6 series of pulse trains produced similar non-degrading current-time profiles characterized by an initial current of 90–120 nA and a saturation around 60 nA. The gradual decrease of the initial current observed from the 1st to the 6th series can be attributed to charge trapping effects and is noticeable only for the first 4–5 readout pulses. Comparison of data collected by sequential pulse readouts of the DW conduction on the PZT surface with the AFM-tip and on the electrode (Fig. 2n) show a very significant difference. The tip-readout method (yellow curve) resulted in data scattering within the range of 15–50 nA in contrast to the measurements by the electrode (red dots), where a smooth curve with scattering of <4% was observed. Additionally, the curve measured using a 2 V DC bias applied through the electrode (blue line) is virtually indistinguishable from the pulse readout data (Fig. 2n). Overall, the data in Fig. 2 demonstrate single-DW-based memristor operations with remarkably high and stable on-state conduction of 50–100 nA at 2 V, using a $250 \times 250$ nm$^2$ electrode (resulting in a 1D-channel current density of 200–400 nA/μm).

## Domain wall conduction at cryogenic temperatures

To further analyze the nature of conduction at the 180°-DWs, a series of scanning probe measurements have been carried out at cryogenic temperatures in ultra-high vacuum (UHV) (Fig. 3). The entire series of scans at temperatures from 4 K to 42 K have been performed using the same conductive diamond probe in order to enhance quantitative comparison. Prior to the poling experiments, pristine areas characterized at 4 K revealed a conduction pattern coinciding with the narrow a-domains, similar to the RT-data and to previously published results[20] (see Supplementary Fig. 4). Because of the UHV environment and the cryogenic temperatures, the voltages required for polarization switching and conduction measurements were higher compared to those used for the RT-measurements shown in Fig. 1. Specifically, the polarization switching required a sample bias of 7 V while the 90°-DW conduction was detectable only above 5 V. For reliable poling we used a sample bias of 9 V, which resulted in uniform square-like poled regions at 4 K (Fig. 3a–c) and 42 K (Fig. 3f–h). As a side effect of using higher poling voltages, we observed some particle deposition on the surface inside the scanned areas (see Supplementary Fig. 5), accompanied with a partial blurring of the cAFM images. Due to the complex conductive patterns observed, all further analyses were therefore focused on zones free of these distorting artifacts. In all cAFM images in Fig. 3 one readily identifies a strong nA-range conductive response of newly created 180°-DWs. These conductive traces persist through sequential scans with varying scanning angle (Fig. 3d, e) or changing bias (Fig. 3i, j). In addition to the 180°-DWs, the 90°-DWs inside the poled region showed an enhanced conductivity of hundreds of pA. Within the poled region, zones with high density of a-domains tend to appear as extended regions of high conduction. I–V curves measured up to 4 V on individual 90°- and 180°-DWs illustrate the typical current response observed at 42 K (Fig. 3k). Analysis of the data acquired at 42 K and 4 K using the same measurement protocol and the same probe revealed their remarkable similarity. Direct comparison of the conduction traces of 180°-DWs measured at 4 K and 42 K with the same sample bias of 4 V show strikingly close quantitative characteristics. Current values - approaching 0.8 nA (Fig. 3l, m) - have been obtained for both temperatures by averaging the conduction data over a 180°-DW segment, chosen in a way to minimize any possible interference with the conductive 90°-DWs. It was possible to control and modify the conductive channels, produced at the cryogenic temperatures, by external biases identical to the RT-measurements and similar probe/PZT interface limited I–V characteristics (polarity dependence and

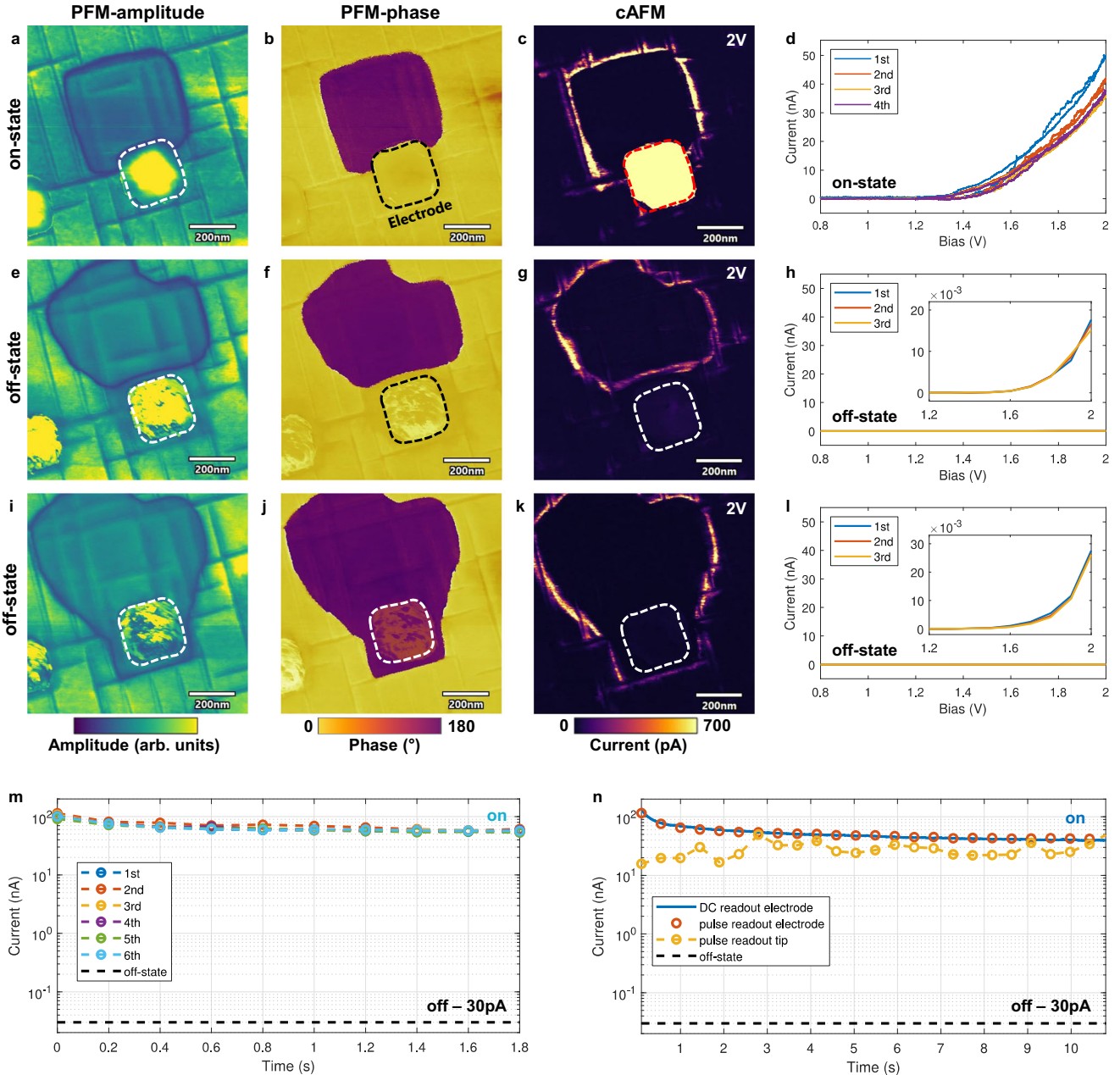

**Fig. 2 | Low-voltage readout of single 180°-DW devices using sub-micron electrodes.** PFM-amplitude (left column), PFM-phase (center) and cAFM (rigth) scans show the capacitor device (outlined by dotted line) connected with a single 180°-DW (on-state) and uniformly poled without a connected domain wall (off-state). cAFM maps were taken with a sample bias of 2 V. **a–d** On-state of the device. The 180°-DW is pushed inside the capacitor area ensuring a stable conduction path between the top and bottom electrode. Consecutive *I–V* curves are performed in **d** by placing the AFM-tip onto the electrode. Stable and low-noise current readouts of 40–50 nA were achieved for <2 V. **e–h** Off-state of the device. After applying to the top electrode a positive voltage pulse exceeding the coercive field value, the 180°-DW is pushed away from the device, breaking the conduction path and inducing the high-resistance state. In **h** consecutive *I–V* scans are performed

through the electrode showing low sub-20 pA currents. **i–l** Off-state of the device. By using a voltage pulse of opposite polarity (negative), the poled region can be extended to fully cover the electrode area (with no domain wall touching the electrode). This results in similar *I–V* characteristics **l** as shown in **h** with only tens of pA currents. **m** Consecutive pulse train measurements performed on an on-state device. Each pulse train consists of 10 triangular voltage pulses of 2 V/200 ms. **n** Comparison between different readout methods. Yellow dotted line indicates a pulse readout by placing the tip on a 180°-DW. The red dotted line shows the same readout scheme with the tip sitting on the electrode. Blue solid line indicates a readout through the electrode but with a constant 2 V DC readout. The black line in **m** and **n** indicates the off-state conductance level of 30 pA.

non-linearity) were observed. The conductive traces could be completely erased by removing the switched domains through backpoling with a sample bias of ≤−5 V.

These results of cryogenic scanning probe microscopy provide a valuable insight into the physical origin of the observed 180°-DW conduction. In particular, they imply an intrinsic nature associated with the domain walls' electronic properties rather than an extrinsic

defect-driven mechanism. The formation of a 2D electron gas, characterized by its metallic-like conduction is consistent with the non thermally-activated conduction observed at the 180°-DWs, which supports remarkably high current levels down to 4 K. With at least 1 nA at 4 V measured by a scanning diamond probe in a cryogenic environment, this illustrates the potential for applications in a wide temperature range.

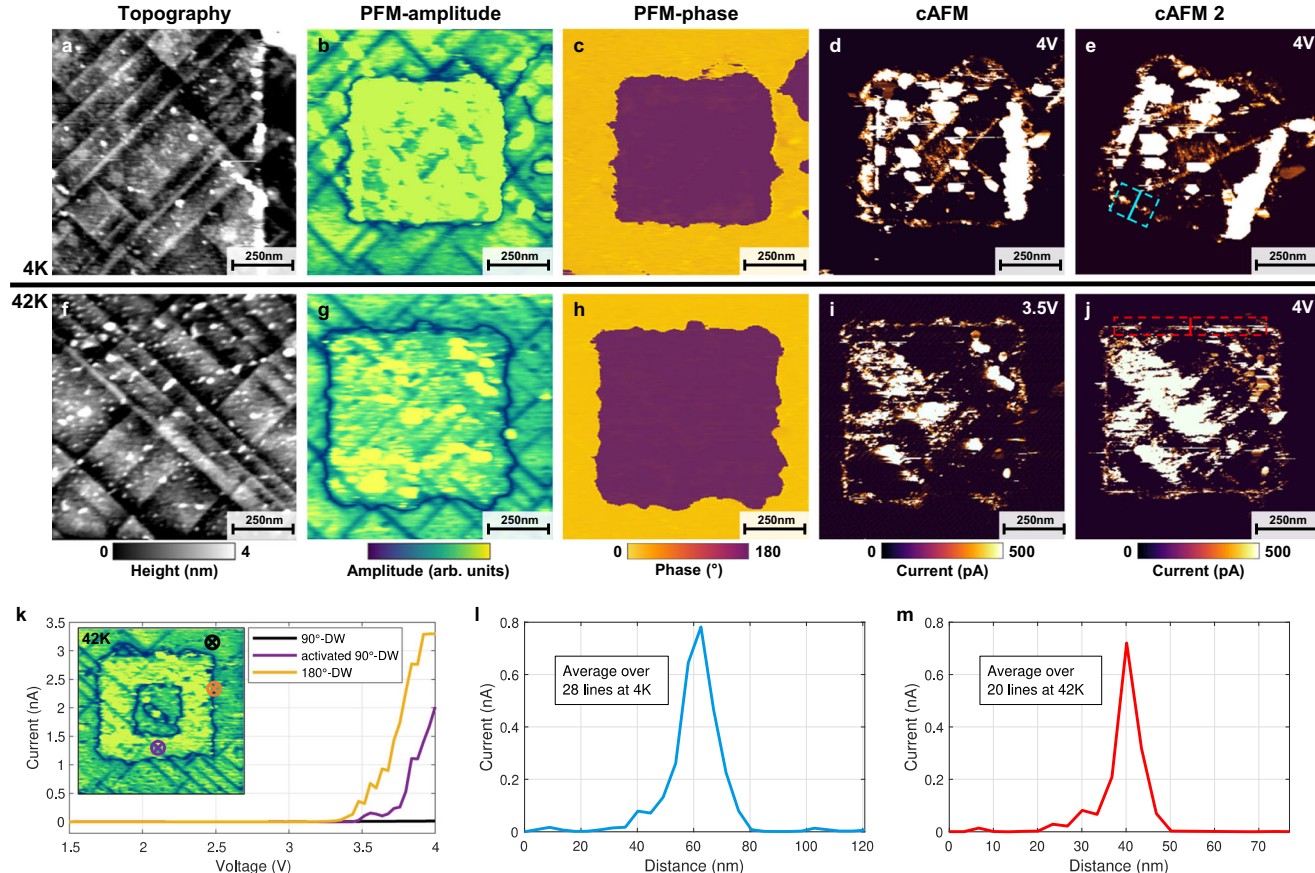

**Fig. 3 | Conduction response of the 180°-DWs at cryogenic temperatures.**
**a–e** Topography, PFM and cAFM images of poled regions at 4 K. **f–j** Same images as
in **a–e** at 42 K. In the PFM-phase images **c** and **h** the clear 180° contrast between
pristine (yellow) and poled (purple) domains is visible. The 180°-DWs (black square
outlines in **b** and **g**), show strong conductivity as confirmed by the cAFM images.
This conduction is detectable under varying imaging parameters like scanning
angle (**d–e**) or voltage bias (**i–j**) both at 4 K and 42 K. In all cAFM images, in addition
to 180°-DWs, an enhanced conduction of the 90°-DWs inside the poled regions is
observed. **k** Shows single-spot *I–V* characteristics at 42 K of a pristine 90°-DW
outside the poled region (black), the enhanced 90°-DW conduction inside (purple)
and the strong 180°-DW response (orange). The insert indicates the positions where
the *I–V* curves where taken with the tip as a top electrode. **l**, **m** An averaged cross-
section profile at 4 K and 42 K collected on the marked areas in **e** and **j** show similar
maximum current around 800 pA for both temperatures.

## Domain wall transport: interplay between a- and c-domain boundaries

A possible interplay between the ferroelastic a-domain and c-domain
boundaries is another issue essential for illucidation of the domain wall
conduction mechanism. Indeed, prior to poling, conductive 90°-DWs
were already observed in the pristine PZT. The data in Fig. 1g–i show
that freshly poled c-domains tend to adopt the form defined by the
a-domain pattern. Furthermore, low-temperature measurements show
an enhanced conductivity of 90°-DWs within the area of reversed
c-domains and similar non thermally-activated conduction character-
istics of the 90°- and 180°-DWs. These observations suggest an inter-
dependence between the transport properties of the two domain wall
types. In this context, it is important to clarify if the conductive 180°-
DWs can exist as stand-alone entities or only in conjunction with
a-domains. The latter scenario appears to take place in the data pre-
sented in Fig. 4a–c. The measurements were carried out according to
the same poling-scheme as in Fig. 1. In the topography and PFM image,
the formation of dense a-domain networks is observed concurrently
with the reversal of the c-domains. Both PFM (Fig. 4b) and cAFM
(Fig. 4c) maps show that at least some segments of the conductive
180°-DW adopt the zigzag shape defined by the new a-domains.
Indeed, in these areas, one can not distinguish the conductive traces of
the 180°-DWs from the conductive response of the 90°-DWs.

Another type of domain wall formation is shown in Fig. 4d–f
where the same poling procedure resulted in a different domain

configuration. It is clear from topography and PFM data (Fig. 4d, e) that
the a-domain density is much lower compared to the case in Fig. 4a, b.
Consequently, long segments of stand-alone 180°-DWs with curved
arbitrary shape could be observed. Consecutive cAFM scans (Fig. 4f)
show that the conduction of such isolated 180°-DWs is comparable
with other segments of the 180°-DW, which are fully aligned and even
overlap with a-domains. Moreover, the high resolution PFM and cAFM
scans (Fig. 4e, f) allow for a separation between the closely spaced 90°-
and 180°-DWs. These measurements demonstrate that despite an
apparent interplay between the two domain wall types, the robust
180°-DW conduction cannot be reduced to a phenomena of activation
of 90°-DWs and that the real scenario is more complicated.

## Phase-field simulations

To gain further insights into the origin of the 180°-DW conductivity
and the interplay between the a- and c-domain boundaries, we per-
formed quantitative phase-field simulations of the poling dynamics.
For this, we model the part of a 50 nm PZT film, with lateral dimensions
of 500 × 500 nm, strained by the substrate with a tensile strain of $u_m$ =
0.35%. The pristine state is prepared from the paraelectric phase by
field-cooling. Interaction of free carriers with the charged domain walls
is accounted for by the introduction of the Thomas-Fermi-Debye-
Hückel screening term into the electrostatic Poisson equation. The
carriers' concentration is accounted for by the screening length $\delta$. The
details of the simulations are given in Methods.

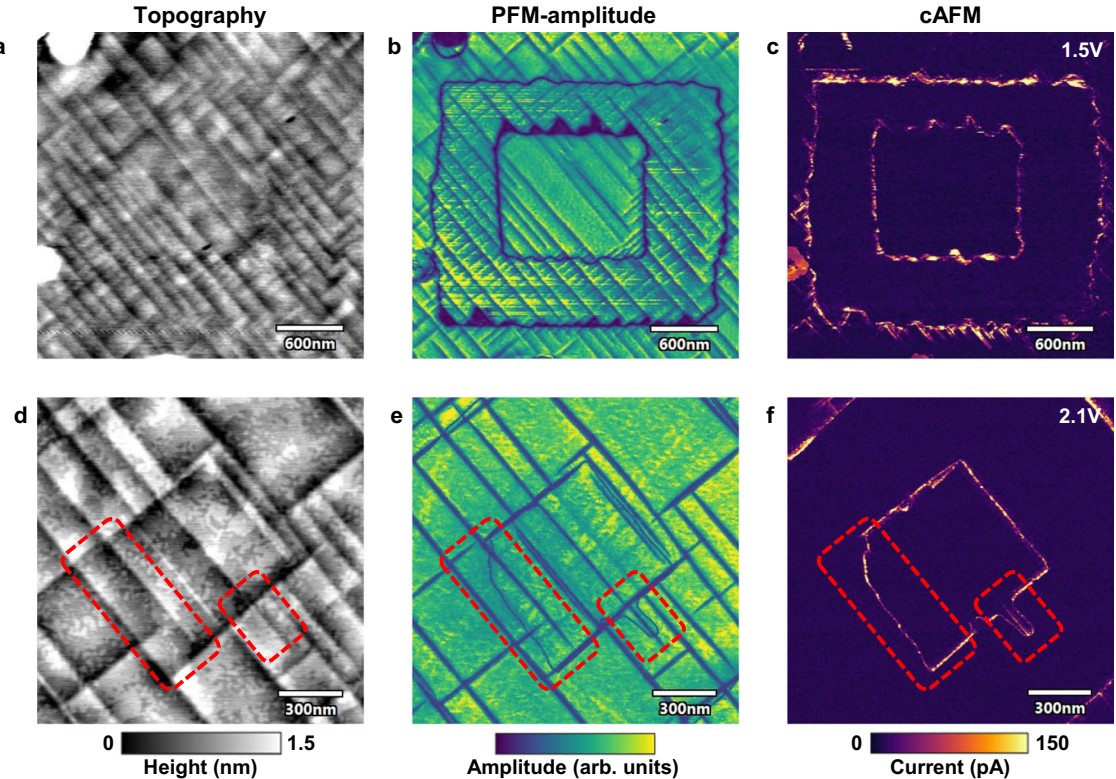

**Fig. 4 | Enhanced resolution topography, PFM and cAFM images of PZT surface with 180°-DWs.** In the two examples (upper and lower row) a square in a square poling done by a AFM-tip produces different domain patterns. **a–c** A peculiar a-domain formation behavior is observed. After poling, a large number of short a-domains form at the position of the 180°-DW and partially overlap with it. This is readily seen in the topography **a** and PFM-amplitude **b** image. From the cAFM image of the same area **c**, a zigzack like conductive trace is recognizable, but it can not be determined if the conduction stems from the 180°-DW or the new short 90°- DWs. **d–f** A zoomed view on closely spaced 180°-DWs and 90°-DWs. In this set of images, it is clear that the 180°-DW exists as a stand-alone object without any associated zigzag-shaped a-domains. The parts marked in red are particularly interesting, here the 180°-DW is readily seen in the PFM image **e** (no topography changes in **d**) with a curved shape and without any ferroelastic a-domains in its close proximity. This stand-alone 180°-DW section produces a conduction response clearly seen in the cAFM image **f**.

Figure 5 demonstrates the dynamics of poling in a circular area (diameter of 400 nm). Three stages of the poling are represented: the pristine sample (Fig. 5a), the polarization state when a tip with −6 V bias is applied (Fig. 5b), and the relaxed state after the tip is retracted (Fig. 5c). The top panel shows the view of the emerging domain walls at the film surface, whereas the bottom panel demonstrates the tomographic structure of the domain walls. To visualize the domain wall structure inside the film (bottom panel), the out-of-plane oriented domains are removed.

In compliance with experimental observations, the pristine sample presents a uniform grid of downward polarized c-domains separated by a rectangular crosshatch pattern of thin a-domains. The internal structure of the system reveals a complex intertwined network of nearly 45° inclined 90°-DWs, that minimizes the interplay of elastic and electrostatic energies of the system.

The biased tip polarizes the sample locally with visible demolition of the a-domains at the film surface (Fig. 5b). The separation region between poled and pristine regions is seen from the outside as a relatively broad domain wall, separating up- and down-polarized domains. However, the detailed examination of the tomography images from inside the sample reveal some intriguing features. First, a-domains do not disappear but lurk in the depths of the film, continuing to minimize the elastic energy from the film-substrate mismatch. Second, and more importantly, contrary to the expectations, the separation boundary between the poled and pristine domain does not propagate through the sample as a single vertically oriented 180°- DW, but is attached to at least a couple of 90°-DWs with narrow a-domains in between. Accordingly, the change of polarization

direction from up to down at the separation boundary occurs as a continuous Néel-like rotation of the polarization vector in a broad near-surface region rather than an abrupt Ising-like jump.

More details on the dynamics of the formation of the poled state are presented in the 2D cross-section simulation of Fig. 6. When the tip bias is applied to the pristine sample (Fig. 6a), an inversely-polarized domain forms at the near-surface region (Fig. 6b). It propagates further inside the film until it reaches the substrate surface (Fig. 6c). After further alignment of the 180°-DWs (Fig. 6d), it reaches the final state (Fig. 6e). We observe that during the dynamical formation of this new domain, additional a-domain stripes are created at the boundaries of the growing c-domain. Once formed, they propagate rapidly and form stripes to the film surface with a nearly 45° inclination, as required by electrostatic compatibility conditions. Interestingly, at lower tip voltages the poled c-domains do not reach the final state (Fig. 6e) but get pinned during the process, accompanied by the formation of terminal a-domain shoots, of octopus-like shapes (Fig. 6b, c).

These poling dynamics give a hint for understanding the conductive traces at the poled region boundaries. Importantly, the boundary between poled and pristine regions, which looks from the surface as a single 180°-DW, has in fact a more complex internal structure. As shown in Fig. 6, it is associated with conducting 90°-DWs, which either emerge from the separation boundary at the surface or nucleate from the 180°-DW inside the sample and propagate downwards to the bottom electrode. These in-plane domains can be nested and further connect with other conducting 90°-DWs in the depth of the film. The model suggests that the conductivity of the 180°-DW

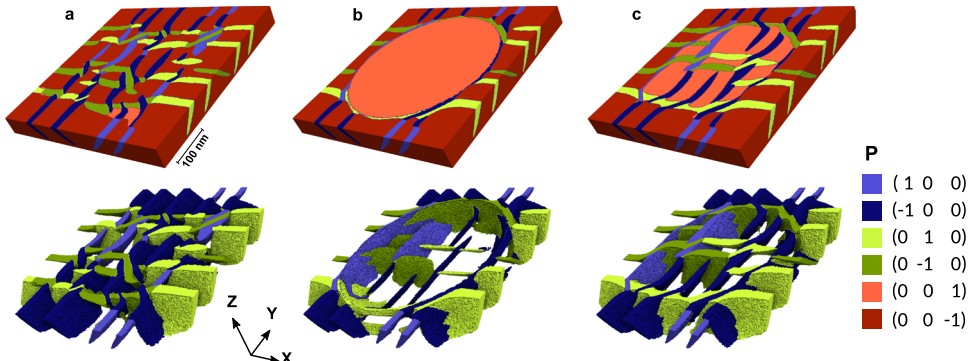

**Fig. 5 | Evolution of the domain pattern in a PZT thin film during poling.** The upper panel shows the distribution of polarization at the surface of the film. The lower panel shows the 3D tomography view of the in-plane domains for which the up- and down-polarized domains are removed. The domains are colored according to their polarization direction, indicated in the legend. **a** Pristine film before poling. **b** Polarization distribution under the bias applied through the tip contacting the surface. **c** Relaxed stage after tip removal. Color legend shows the polarization direction along the crystal axis.

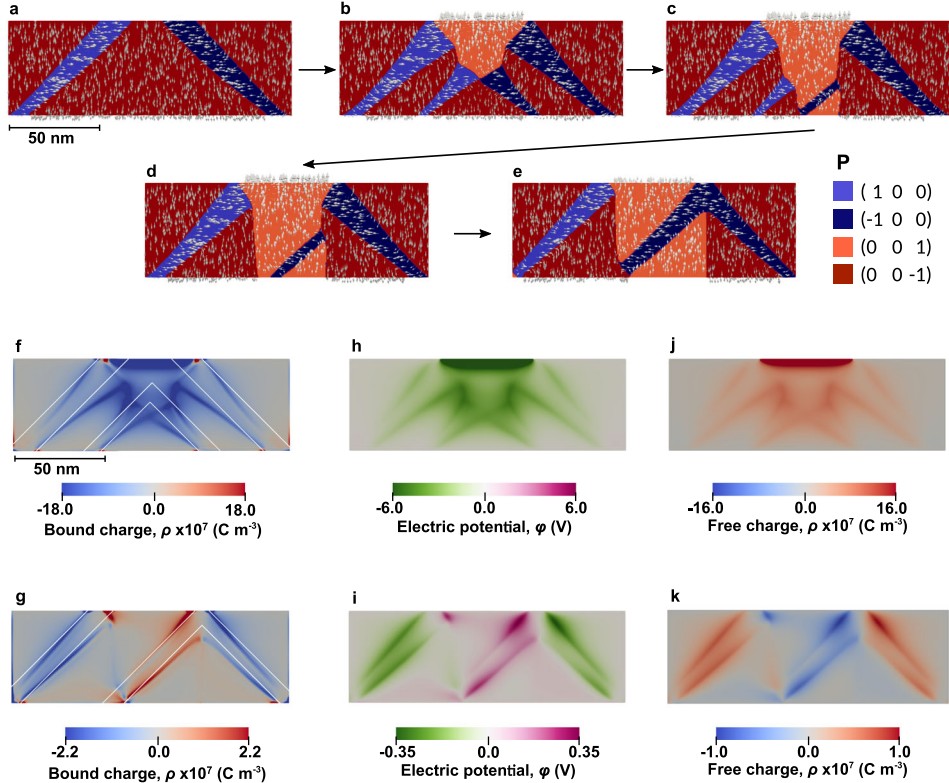

**Fig. 6 | Poling dynamics and electrostatics of domain boundaries. a** Distribution of polarization and domains in pristine PZT prior to poling. **b** The initial poling stage: after the tip contacts the uniformly poled PZT, a new c-domain forms in a near-surface region producing additional a-domain stripes propagating towards the bottom interface. **c** The inversely-polarized c-domain penetrates in depth of the film, reaching the bottom interface. **d** Further evolution of the system leads to an alignment of the 180°-DW with the crystal axis. **e** Relaxed state of the system after tip removal. **f, g** Bound charges concentrated near the bent 90°-DWs for the poling stages **b** and **e**, respectively. The white lines show the charge-neutral direction of the domain walls. **h, i** Electrostatic potential distribution for the same stages of poling. **j, k** Free conductive charges distribution that screens the bound charges for the same stages of poling.

mimic the conductivity of the associated 90°-DWs, which emerge at the poled region boundary.

The origin of the conductivity of the 90°-DWs on DSO-deposited PZT films was shown[20] to originate from the progressive bending of the domain walls from the charge-neutral head-to-tail (HT) or tail-to-head (TH) 45° orientation. The domain wall bending breaks the polarization continuity, resulting in the emergence of bound charges at the domain wall. The now charged domain walls attract the carriers from the bulk of the film, which partially screens the bound charges. The bulk carriers are normally localized at the impurities energy levels, for instance

at oxygen vacancies. Importantly, the potential drop at charged domain walls ~0.8 eV is below the defect level (located at ~−0.6 eV with respect to the bulk conduction band), thus pushing the conduction band at the domain wall region below the Fermi-level and providing the metallic-like conductivity of a 2D electron gas.

Figure 6f–k presents the details of the charge and potential distribution along the domain walls during the poling process (corresponding to Fig. 6b) and after the tip removal (corresponding to Fig. 6e), obtained according to the calculations given in Methods. The density of polarization bound charges within the bent domain walls is

shown in Fig. 6f and g for the respective poling stages. The charge-neutral 45° orientation is indicated by white lines. Figure 6h and i show the corresponding distribution of the charge-induced electrostatic potential that is partially screened in the vicinity of the domain walls. Figure 6j and k demonstrate the distribution of the free charges accumulated in the vicinity of the 90°-DWs, which are responsible for the metallic-like conduction.

In conclusion, we report controllable 2D conductive channels with a remarkably high current density of 200–400 nA/μm at voltages ≤2 V, associated with 180°-DWs in tetragonal PZT. These channels exhibit a metallic-like behavior with non thermally-activated conduction persisting down to 4 K. Individual domain walls could reproducibly be contacted by nanometer-size electrodes forming single-DW-based memristors with time- and readout-stable conductance states characterized by on-state currents of 50 nA and on/off ratios up to $10^3$. Phase-field simulations reveal rather complex domain arrangements behind the apparent single 180°-DWs detected by the surface-based AFM-techniques. The analysis of poling dynamics suggests the formation of interconnected structures of 90°-domains emerging from the 180°-DW. The deviation from the neutral 45° angle of these a-domain networks, entails the formation of partially charged domain walls. This results in a 2D electron gas formation responsible for the extraordinary transport properties observed at the 180°-DWs. This mechanism is different from the previously reported defect-assisted conduction in 180°-DWs in PZT and leads to fundamentally different conduction properties. Thus, the longstanding issues of domain wall electronics relating to low current densities and high operation voltages can be overcome by using highly stable 180°-DWs through accessible domain reconfigurations in one of the most used ferroelectrics, PZT.

## Methods

### Film growth
For this study, highly tetragonal $Pb(Zr_{0.1}Ti_{0.9})O_3$ (PZT) was grown on a (110) $DyScO_3$ (DSO) substrate by pulsed laser deposition (PLD). The DSO substrates with a miscut angle of 0.1° towards the [1-10] direction were acquired from CrysTec GmbH. A thin layer of 20 nm $SrRuO_3$ (SRO) was deposited by PLD before the PZT growth and served as a bottom electrode. The PLD deposition parameters were the following: a 248 nm laser was used with an energy density of $1 J/cm^2$. For SRO and PZT deposition, the substrate temperature/oxygen pressure were 625°C/0.145 mbar and 575°C/0.25 mbar, respectively. The cooling after the film deposition was performed at the controlled rate of 15°C/min with 1 mbar oxygen pressure. XRD-analysis yielded an epitaxial structure and previously performed TEM analysis of similar 60 nm PZT films revealed the bent 45° narrow ferroelastic a-domains of 10–12 nm widths. These a-domains form due to the film/substrate lattice mismatch to minimize the mechanical energy[29]. Their width and density depends on the exact lattice mismatch and results in the formation of a crosshatch pattern[27,30] clearly visible in the topography AFM (height change due to the 90° crystal rotation) and vertical PFM images (drop in amplitude signal due to the in-plane polarization). The polarization in all as-grown c-domains was oriented from the top to bottom interface, as confirmed by the switching dynamics through piezo-force microscopy measurements.

### Topography, PFM and cAFM imaging
All room temperature scanning probe microscopy experiments were preformed using an Asylum Research (Oxford Instruments) Cypher AFM system (Asylum Research Cypher ES) equipped with the environmental scanner. The measurements were carried out in a temperature controlled (306 ± 0.1 K) environmental chamber under a small continuous $N_2$ flux (5–10 mbarg) and were preceded by a 15 min exposure to 200°C in order to eliminate adsorbed water from the surface. For all RT-experiments stiff (40 N/m) conductive boron doped

diamond coated tips from ADAMA Innovations were used: AD-40-AS with standard sharpness (tip radius: 10 ± 5 nm) and AD-40-SS for high resolution images (tip radius: <5 nm). For the PFM as well as cAFM measurements, a Cypher Dual Gain ORCA Holder (capable of measuring between 1 pA and 10 μA) was used. Because of the circuitry of the ORCA holder all biases to the sample are applied with respect to the bottom SRO electrode and the tip (top electrode) was kept at virtual 0 V. All electrical characterizations (I–V curves and pulse measurements) were performed with the AFM by using the tip (or through contacting the top electrode with the tip). For cAFM and PFM images typical scan parameters were as follows: between 1 and 2 Hz scan speed, 0.1–0.2 V deflection setpoint and 20–40 gain factor. PFM images of Figs. 1 and 4 were taken using Cyphers built in contact resonance technique DART to enhance the PFM signal and allow for faster imaging. All PFM data taken on the electrode measurement (Fig. 2) were done using the standard single frequency PFM technique.

### Electrode patterning
Top Cr/Au (thicknesses: 5/20 nm) electrodes with different sizes between 80 nm and 5 μm were deposited using electron-beam lithography (Raith EBPG5000) and metal evaporation (Alliance-Concept EVA 760) together with lift-off techniques.

### Cryogenic AFM-setup
The measurements at cryogenic temperatures down to 4 K were done using an UHV (<$10^{-9}$ mbar) Cryo-SFM, manufactured by Omicron Nanotechnology (currently Scienta Omicron). Conductive diamond coated probes (NaDiaProbes) with a nominal spring constant of 5 ± 1 N/m were used in order to minimize tip degradation during multiple scans. All cryogenic PFM and cAFM scans shown in the present work were done using the same probe.

### Functional
Numerical simulations of domain walls in the ferroelectric thin film are based on the minimization of the Ginzburg-Landau-Devonshire free-energy functional[31] for the pseudocubic ferroelectric material in which the elastic and electrostatic effects are included:

$$F = \int \left( \left[ a_i(T)P_i^2 + a_{ij}P_i^2P_j^2 + a_{ijk}P_i^2P_j^2P_k^2 \right]_{i \le j \le k} + \frac{1}{2}G_{ijkl}(\partial_i P_j)(\partial_k P_l) \right.$$
$$\left. - \frac{1}{2}\varepsilon_0\varepsilon_b[(\nabla\varphi)^2 + \varphi^2\delta^{-2}] + (\partial_i\varphi)P_i + \frac{1}{2}C_{ijkl}u_{ij}u_{kl} - C_{ijkl}Q_{klmn}u_{ij}P_mP_n \right) d^3r$$

$$(1)$$

Here we assume the tensor summation over the repetitive indices that takes the cartesian components $x, y, z$ (or 1,2,3).

Functional (1) comprises the Ginzburg-Landa energy[32] given in the first square brackets. The second term is the polarization gradient energy[33]. Third and fourth terms represent the electrostatic energy, accounting also the screening effects[34]. The last two terms correspond to the elastic energy. The electrostatic potential and strain tensor are denoted as $\varphi$ and $u_{ij}$ respectively. The value of the vacuum permittivity $\varepsilon_0$ is $8.85 \times 10^{-12}$ $CV^{-1}m^{-1}$ and the value of the background dielectric constant $\varepsilon_b$ is 10[35]. The numerical values of the Ginzburg-Landau expansion coefficients $a_{ijk}$, gradient energy coefficients $G_{ijkl}$, elastic stiffness tensor $C_{ijkl}$ and tensor of electrostrictive coefficients $Q_{ijkl}$ are given below.

The electrostatic properties of the system are described by the Poisson equation, $\varepsilon_0\varepsilon_b\nabla^2\varphi = -(\rho_{bound} + \rho_{free})$, that is governed by two types of charges. The density of the bound charges, $\rho_{bound} = -\nabla \cdot \mathbf{P}_{tot}$, is provided by the non-uniform distribution of the total polarization in the bent 90°-DWs, which includes the spontaneous and field-induced parts: $\mathbf{P}_{tot} = \mathbf{P} + (\varepsilon_b - 1)\nabla\varphi$. Here $\varepsilon_0$ is the vacuum dielectric permittivity and $\varepsilon_b \approx 10$ is the background dielectric constant of non-polar ions[35]. The density of the uncompensated free charges is given by the

linearized Thomas-Fermi equation $\rho_{\text{free}} = -(\varepsilon_0\varepsilon_b\delta^2)^{-1}\varphi$. The screening length $\delta$ can be estimated[36] through the Bohr radius $a_0 = 0.053$ nm and the concentration of carriers $n_0 \approx 10^{20}$ m$^3$ as $\delta \approx (a_0/4n_0^{1/3})^{1/2} \approx 1.6$ nm. We use this value for the phase-field calculations.

The distribution of the electrostatic potential $\varphi$ and the elastic strains $u_{ij}$ is found from the respective electrostatic (with screening) and elastic equations:

$$\varepsilon_0\varepsilon_b[\nabla^2 - \delta^{-2}]\varphi = \partial_i P_i \tag{2}$$

$$C_{ijkl}\partial_j(u_{kl} - Q_{klmn}P_m P_n) = 0 \tag{3}$$

## Material coefficients

The coefficients of the Ginzburg-Landau expansion for Pb(Zr$_{0.1}$Ti$_{0.9}$)O$_3$ at room temperature[37] are as follows: $a_1 = -0.1618 \times 10^5$ C$^2$m$^2$N, $a_{11} = 0.3883 \times 10^9$ C$^{-4}$m$^6$N, $a_{12} = 0.6357 \times 10^9$ C$^{-4}$m$^6$N, $a_{111} = 0.2518 \times 10^9$ C$^{-6}$m$^{10}$N, $a_{112} = 0.8099 \times 10^9$ C$^{-6}$m$^{10}$N, $a_{123} = -4.3588 \times 10^9$ C$^{-6}$m$^{10}$N (the second order coefficients $a_{ij}$ are taken for the zero-strained sample. They are calculated from the stress-free coefficients, using the standard procedure[38]). The values of the electrostrictive tensor coefficients are: $Q_{1111} = 0.085$ C$^{-2}$m$^4$, $Q_{1122} = -0.0251$ C$^{-2}$m$^4$, $Q_{1212} = 0.0328$ C$^{-2}$m$^4$ [37]. Components of the elastic stiffness are: $C_{1111} = 1.7 \times 10^{11}$ m$^{-2}$N, $C_{1122} = 0.76 \times 10^{11}$ m$^{-2}$N, $C_{1212} = 0.83 \times 10^{11}$ m$^2$N. The gradient energy coefficients[39] are: $G_{1111} = 2.77 \times 10^{-10}$ C$^{-2}$m$^4$N, $G_{1122} = 0$, $G_{1212} = 1.38 \times 10^{-10}$ bC$^{-2}$m$^4$N.

## Phase-field modeling

The non-linear differential relaxation equation is used to find the minima of the free energy (1):

$$-\gamma\frac{\partial P}{\partial t} = \frac{\delta F}{\delta P} \tag{4}$$

Here $\gamma$ is a time-scale parameter which is taken to be equal unity. The non-linear part of the equations is accompanied by two linear systems of equations defined by the screened Poisson Eq. (2) and the equation of linear elasticity (3).

The phase-field simulations were conducted with the help of the FEniCS software package[40]. Two- and three-dimensional rectangular computational regions are represented by structured triangular and tetrahedral finite element meshes, respectively, that were created with the 3D mesh generator *gmsh*[41]. The solutions for **P**, $\varphi$ and $u_{ij}$ was sought in the functional space of piece-wise linear polynomials.

The initial quenching from the paraelectric state was conducted with the imposition of Dirichlet boundary conditions at the bottom side of the computational region $\varphi_{\text{bot}} = 0$ and $\varphi_{\text{top}} = 1 \times 10^{-6}$ V at the top side. The application of the tip was simulated by imposition of the Dirichlet boundary condition $\varphi = -6$ V in the circular area of the tip application and zero everywhere else at the top side of the computational region. Substrate-induced strain is taken into account by imposition of the Dirichlet boundary conditions on components of the displacement vector **u** at the bottom surface of the thin film, $u_x = u_0 x/L_x$ and $u_y = u_0 y/L_y$, where $u_0 = 0.35\%$ is the strain value, $L_x = 500$ nm and $L_y = 500$ nm are in-plane geometrical dimensions of thin film for the 3D case (as in Fig. 5) and $L_x = 150$ nm for the 2D case (as in Fig. 6). Variables **P** and $\varphi$ are constrained with periodic boundary conditions in the $x$ and $y$ directions.

The approximation of the time derivative on the left-hand side of Eq. (4) is accomplished by BDF2 variable time stepper[42]. The initial condition for polarization at the first-time step is a random distribution of the polarization vector components in the range of $-10^{-6}$ to $10^{-6}$ Cm$^{-2}$. Newton method with line search is used to solve the non-linear system arising from Eq. (4). To solve the linear system on each non-linear iteration and systems defined by Eqs. (2) and (3), the generalized minimal residual method with restart is used[43,44].

## Reporting summary

Further information on research design is available in the Nature Portfolio Reporting Summary linked to this article.

## Data availability

The data generated and analyzed during this study is available from the corresponding author upon reasonable request.

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

## Acknowledgements
This project has received funding from the EU Horizon 2020 program under the MSCA-ITN action MANIC, grant agreement No 861153 (F.R., I.S., Y.T., I.L.) and MSCA-RISE action MELON, grant agreement No 872631 (Y.T., I.L.). The authors gratefully acknowledge the help of Dr. Barbara Fraygola for PLD deposition, and Dr. Kanghyun Chu for UHV cryogenic AFM support.

## Author contributions
F.R., I.S. and A.I. conceived the project. F.R. and I.S. carried out the experimental work and analyzed the data. Y.T. and I.L. performed the theoretical work and carried out the simulations. All authors contributed to the preparation of the manuscript.

## Competing interests
The authors declare no competing interests.
