## [Peer Review File · Nature Communications]

Giant switchable metallic-like conduction in 180° domain walls in tetragonal Pb(Zr,Ti)O₃REVIEWER COMMENTS

Reviewer #1 (Remarks to the Author):

In this manuscript, Risch et. al. reported switchable metallic-like conduction in the 180° domain walls of Pb(Zr,Ti)O₃. cAFM measurements and phase-field modeling are performed to understand the domain arrangements and switchable conductivity. These results demonstrated the switchable conductivity in the 180° domain walls of Pb(Zr,Ti)O₃. However, evidence for the claimed “metallic-like behavior” is not sufficient. After providing more experiment evidences for the metallic-like conductivity and completing the discussion of the following issues, I can recommend this manuscript for publication in Nature communications.

1. “metallic-like behavior” is claimed in this work, however, I-V curves show a semiconducting behavior. The author should provide more evidence to support the claimed “metallic-like behavior”.
2. The authors should compare it with other reported works to show this work is “Giant switchable”. For example, comparing the switching conductivity/changes with numbers from other works.
3. In Fig. 2m and 2n, the current values for off status should be added to the black line with dots.
4. Fig. 1d shows a blue line of about 2500nm, however, the data shown in Fig. 1e is just 0-1500nm. The author should explain or correct this.
5. For comparison, the color for Fig. 4 a-j should use the same color themes as Fig. 2.
6. The scale bar is missing for Fig. 3b-e and Fig. 3g-i.

Reviewer #2 (Remarks to the Author):

As in the recent article (titled nonvolatile ferroelectric domain wall memory integrated on silicon) published in NATURE COMMUNICATIONS, ferroelectric domain walls have exhibited many novel properties, including metal conductance, magnetoresistance and photovoltaic effect, as nanoscale transition regions for domain separation in ferroelectric materials. Based on the erasable control of its nanoscale size and conductivity, ferroelectric domain walls have important application prospects in high-density, low-energy-consumption non-volatile memory.

The author's research on conductive domain walls (DWS) in ferroelectrics is very interesting. I have some doubts:

1. The author mentioned "... Ferroelastic 90 ° - domains and 180 ° - domains." in the abstract. The author did not explain how to distinguish ferroelectric domains from ferroelastic domains in the samples. Please explain.
2. The author mentioned "The PZT film showed predominantly downward-oriented c-domains interrupted by thin 10 nm wide ferroelastic a-domains formed due to the specific strain conditions as described in." What is the domain wall thickness in the sample? Is there a clear measurement result?
3. The author mentioned "As seen in the AFM topography and PFM images (Fig.1a-c), these in-plane a-domains form a rectangular crosshatch pattern separating the uniformly polarized c-domains." How are in-plane a-domains and out-of-plane c-domains determined?
4. The author mentioned "After poling, the switched c-domains slightly shrink down over time

and reach their stable configuration by adopting - wherever possible - the boundaries defined by the ferroelastic a-domains (Fig. 1g-i)." How are ferroelectric domains and ferroelastic domains distinguished in the samples? Question 4 is the same as question 1.

5. The author mentioned "The individual domain walls show non thermallyactivated metallic-like conduction and support remarkably high current levels down to 4 K (at least 1 nA at 4V measured by a scanning diamond probe with 10 nm radius), ..." What is the domain wall thickness in the samples? Is there a clear measurement result? Can some domain walls be marked in the figure?

6. The author mentioned "A possible interplay between the ferroelastic a-domain and c-domain boundaries is another issue essential for illucidation of the domain wall conduction mechanism." How to determine that ferroelastic domains rather than ferroelectric domains exist in the samples? Ferroelastic domains and ferroelectric domains do not co-exist?

7. The author mentioned "This results in a 2D electron gas formation responsible for the extraordinary transport properties observed at the 180° -DWs." in the conclusions. The author should explain the two-dimensional electron gas in the corresponding part of the manuscript.

Reviewer #3 (Remarks to the Author):

The authors studied a classical ferroelectric system and found the created 180 domain walls in the a/c domains show giant conduction and designed a device based on this effect. These results are very interesting. I think this manuscript could be published after addressing these issues.

1. I doubt the metallic-like conduction claimed by the authors. In Fig. 1o, the current is about 90 nA at the bias of 2 V at the room temperature. In Fig. 3k, the current is only about 3 nA at the bias of 4 V at 42 K. Comparing these two values, one could see that the conductivity increase with the increase of temperature. In other words, it behaves like a semiconductor.

2. In the cAFM images at the cryogenic temperatures, there are many bright spots. The authors attribute them to some deposited particles. Why they disappear at the room temperature?

3. The authors say little about the technical details of Fig. 6. Is it a cross section of Fig. 5? Or is it a new simulation (maybe 2D)?

Besides these issues, there are some minor problems.

Page. 3: "the progressive bending of the domain walls from the charge-neutral head-to-head (HH) or tail-to-tail (TT) 45° orientation". The charge-neutral domain walls should be the head-to-tail type.

Page. 3: "charged-induced electrostatic potential". It should be "charge".

Referees' response letter for the manuscript:

Giant switchable non thermally-activated conduction in 180° domain walls in tetragonal Pb(Zr,Ti)O₃

Felix Risch¹, Yuri Tikhonov², Igor Lukyanchuk², Adrian M. Ionescu¹, Igor Stolichnov^{1*}

¹Nanoelectronic Devices Laboratory, Ecole Polytechnique Federale de Lausanne (EPFL), CH-1015, Switzerland

²University of Picardie, Laboratory of Condensed Matter Physics, Amiens, 80039, France

*The correspondence should be sent to igor.stolitchnov@epfl.ch

Dear Referees,

we are grateful for your careful reading of the manuscript, helpful comments, and the identification of shortcomings in our presentation. We carefully analyzed all referees' points and used them when improving the manuscript, which we are now resubmitting.

Below we address each concern point by point.

Sincerely,

Felix Risch (in the name of all co-authors)

Reviewer 1

"In this manuscript, Risch et. al. reported switchable metallic-like conduction in the 180° domain walls of Pb(Zr,Ti)O₃. cAFM measurements and phase-field modeling are performed to understand the domain arrangements and switchable conductivity. These results demonstrated the switchable conductivity in the 180° domain walls of Pb(Zr,Ti)O₃. However, evidence for the claimed "metallic-like behavior" is not sufficient. After providing more experiment evidences for the metallic-like conductivity and completing the discussion of the following issues, I can recommend this manuscript for publication in Nature communications."

Authors' response: We agree with the reviewers' assessment that the statement of metallic-like conduction behavior needs clarification. Below we address the specific questions/comments and relevant changes made in the manuscript and supplementary information.

Comments

1. *"metallic-like behavior" is claimed in this work, however, I-V curves show a semiconducting behavior. The author should provide more evidence to support the claimed "metallic-like behavior."*

Answer

This comment together with the remarks from reviewer 3 have clearly exposed to us the need for a deeper discussion of the physics behind the measured DW conduction. Our analysis suggests a metallic DW transport, however the measured current response does not show a linear I-V curve. The reason is that the transport is limited by the potential barrier at the interface between the probe (or electrode) and the PZT. Within this model the resistance of this barrier is significantly higher compared to the 2D electron gas at the DW, therefore the resulting I-V curve is mainly determined by this potential barrier. The electron injection from the probe to the PZT occurs in the tunneling regime, similar to the measurements earlier reported in Ref. 1, which is consistent with our experimental observations:

- the measured conduction shows a weak temperature dependence (Fig. 3 from the manuscript and newly measured data discussed below and presented in Supplementary Figure 4)
- the I-V curves are polarity-dependent (Figure 2 in Supplementary information), a significant current is observed only for the positive sample bias
- classic Fowler-Nordheim (FN) formalism provides an adequate description of the measured I-V curve (FN fit is added to the revised Fig. 1 and described in detail in Supplementary Note 5)

The reason for us to use the term "metallic-like conduction" is that the robust DW conduction was still measured down to 4K, which is not consistent with the thermally-activated conduction occurring in semiconductors. In our opinion, our demonstration of very similar conduction for the measurements at temperatures – varying by factor >10 – from 4K to 42K (as shown in Fig. 3) is a striking evidence, supporting a metallic-like 2D-electron-gas type transport at the 180° DWs.

On the other hand, the metallic conduction could not be evaluated directly in the configuration of this experiment because of the interface barrier. Further proofs would

require a planar device on an isolating substrate. Achieving similar DW properties in such a system is a technological challenge, which is beyond the scope of this work.

In this context, we understand the concern of the reviewer regarding the use of the term "metallic conduction". In order to comply with this comment and better reflect the essence of our experimental finding we decided to modify the title replacing the term "metallic" by "non thermally-activated": **"Giant switchable non thermally-activated conduction in 180° domain walls in tetragonal Pb(Zr,Ti)O₃".**

In response to the request to provide more experimental results we have added to the supplementary information (Supplementary Fig. 4) a temperature dependent conduction measurement on 180°-DWs. The measurements were carried out using the environmental control Cypher-AFM scanner in the temperature range of 35°C-100°C. The observed general trend is a decrease of DW conduction vs temperature, which can be partially explained by a polarization decrease and an increase of the effective barrier thickness (2D gas retracts deeper away from the surface). All in all, it is very clear that the measured conductivity is non thermally-activated as suggested in the manuscript.

2. *"The authors should compare it with other reported works to show this work is "Giant switchable". For example, comparing the switching conductivity/changes with numbers from other works."*

Answer

In Supplementary Table 1, we have included a table summarizing relevant DW conduction data. The reported values of conduction of stable uncharged DWs are within pA range and typically require higher voltage compared to our measurements. In this context, the DW conduction in stable nominally uncharged 180°-DWs that reaches 100nA at 2V is termed „Giant“ in our manuscript. Comparable levels of current were achieved only for charged domain walls, which form transiently under voltage and require special poling procedures.

3. *"In Fig. 2m and 2n, the current values for off status should be added to the black line with dots."*

Answer

The Figure has been updated accordingly.

4. *"Fig. 1d shows a blue line of about 2500nm, however, the data shown in Fig. 1e is just 0-1500nm. The author should explain or correct this."*

Answer

Thank you for noticing, we have adjusted the indication line and placed it to the right position.

5. *"For comparison, the color for Fig. 4 a-j should use the same color themes as Fig. 2."*

Answer

This has been changed. The different color themes resulted from the different scanning probe systems, and we failed to fix the color scheme in the original manuscript.

6. *"The scale bar is missing for Fig. 3b-e and Fig. 3g-i."*

Answer

Thank you for noticing, the missing scale bars have been added.

Reviewer 2

„As in the recent article (titled nonvolatile ferroelectric domain wall memory integrated on silicon) published in NATURE COMMUNICATIONS, ferroelectric domain walls have exhibited many novel properties, including metal conductance, magnetoresistance and photovoltaic effect, as nanoscale transition regions for domain separation in ferroelectric materials. Based on the erasable control of its nanoscale size and conductivity, ferroelectric domain walls have important application prospects in high-density, low-energy-consumption non-volatile memory. The author's research on conductive domain walls (DWS) in ferroelectrics is very interesting. I have some doubts:”

Authors' response: We are grateful to the reviewer for drawing our attention to this new important paper. We have included it in the state of the art overview and referred to it in the new version of our manuscript. In the following we will comment on the reviewers' concerns.

Comments

1. *„The author mentioned "... Ferroelastic 90 ° - domains and 180 ° - domains." in the abstract. The author did not explain how to distinguish ferroelectric domains from ferroelastic domains in the samples. Please explain.”*

Answer

The choice of epitaxial highly tetragonal PZT on SRO/DSO presents an advantage of a well-studied domain structure that has been thoroughly characterized and documented in previous publications. We understand the concern of the reviewer about the lack of information about the domain structure and added these information in the film growth section in "methods" together with corresponding references.

The domain structure in the studied system consists mainly of ferroelectric domains with polarization parallel to the (001) direction normal to the film/substrate surface (c-domains). The lattice parameter of the tetragonal unit cell in this orientation is closely compatible with the substrate lattice parameter. These domains are not ferroelastic i.e., switching of the domains between the two opposite directions along the [001] axis do not change the mechanical conditions of the film (hence no measurable topographic changes between the switched c-domains as confirmed in Fig.1). These domains can be switched by the electric field and their boundaries are vertical and hence neutral.

Because of the imperfect matching of lattice parameters between the film and substrate, ferroelastic domains are spontaneously formed in order to minimize the mechanical energy. In the ferroelastic domains of the studied PZT the polarization is oriented in-plane (a-domains), therefore the lattice mismatch and strain can be minimized via a suitable configuration of the a-domains. Their width and density depend on the lattice mismatch as analyzed theoretically and proven experimentally in previous studies (Refs 2,3,4). The ferroelastic domains are easily distinguished because of their stripe geometry and orientation defined by the principle crystallographic directions (resulting in the formation of cross-hatch patterns). These ferroelastic domains are sensitive to mechanical stress and can be altered by local pressure e.g. using the AFM probe.

To summarize the answer on how to distinguish ferroelastic from ferroelectric domains, the ferroelastic domains can be readily recognized in AFM/PFM images. First, they are visible as topographic changes in the AFM images (showing a cross-hatch pattern,

unlike c-domains) and second, their typical stripe geometry seen in PFM images follows the cross-hatch pattern from the topographic images. In contrast to that, ferroelectric c-domains have an arbitrary shape that depends on the poling conditions and show no topographic changes (as demonstrated in Fig.1). Additionally, compared to the ferroelectric c-domains, ferroelastic a-domains show a low vertical PFM-amplitude signal as expected from the in-plane domains.

2. „The author mentioned "The PZT film showed predominantly downward-oriented c-domains interrupted by thin 10 nm wide ferroelastic a-domains formed due to the specific strain conditions as described in." What is the domain wall thickness in the sample? Is there a clear measurement result?"

Answer

The thickness of the entire a-domains in similar PZT/SRO/DSO samples has been directly measured by TEM in (Ref. 3,4,5) and was found to be about 10-15nm. The thickness of the DW is much more difficult to determine, however STEM-HAADF analysis yields a thickness of about 1nm (Ref. 5), which is in agreement with the theoretical estimates (Ref. 6). Scanning probe techniques (cAFM, PFM, etc.) do not provide a sufficient resolution, which is mainly limited by the tip radius of several nanometers, to resolve the real width of the DWs.

3. "The author mentioned "As seen in the AFM topography and PFM images (Fig. 1a-c), these in-plane a-domains form a rectangular crosshatch pattern separating the uniformly polarized c-domains." How are in-plane a-domains and out-of-plane c-domains determined?"

Answer

As discussed in our answer to the comment 1, the ferroelastic a-domains are easily distinguishable from c-domains.

First, a-domains are clearly seen in the topography images because the lattice parameters « a » and « c » are different. In contrast, the ferroelectric c-domains are invisible in the topography images. Additionally, they can be distinguished in the PFM data because of the difference in their amplitude of piezoelectric response. The stripe-like geometry itself is a feature that helps recognizing ferroelastic a-domains, in agreement with the established theory of domain structures (Ref. 3).

4. "The author mentioned "After poling, the switched c-domains slightly shrink down over time and reach their stable configuration by adopting - wherever possible - the boundaries defined by the ferroelastic a-domains (Fig. 1g-i)." How are ferroelectric domains and ferroelastic domains distinguished in the samples? Question 4 is the same as question 1."

Answer

As explained in detail in the answer to comment 1, the vertical PFM scans from Fig. 1 show a map of c-domains clearly defined by their amplitude and phase and their possibility to be switched by an electric field. The a-domains are seen in the corresponding topography images and in the PFM scans as minima of the amplitude signal. Because the domains of both types are clearly detected and distinguishable within the same image, one easily recognizes the boundaries of c-domains coinciding with the a-domains.

5. *"The author mentioned "The individual domain walls show non thermally activated metallic-like conduction and support remarkably high current levels down to 4 K (at least 1 nA at 4V measured by a scanning diamond probe with 10 nm radius), ..." What is the domain wall thickness in the samples? Is there a clear measurement result? Can some domain walls be marked in the figure?"*

Answer

As discussed in the answer to comment 2, the STEM data from an identically fabricated sample yield an a-domain boundary thickness of about 1nm (Ref. 5,6). The domain walls cannot be marked in the figure with such precision because the AFM resolution is about 5-10nm (limited by the tip sharpness). Therefore, the domain boundary positions can be located in the PFM maps (minima in the amplitude signal, as discussed), but can not determine their absolute thickness.

6. *"The author mentioned "A possible interplay between the ferroelastic a-domain and c-domain boundaries is another issue essential for illucidation of the domain wall conduction mechanism." How to determine that ferroelastic domains rather than ferroelectric domains exist in the samples? Ferroelastic domains and ferroelectric domains do not co-exist?"*

Answer

As discussed in our answers to the previous comments ferroelastic and ferroelectric domains do coexist; they are concurrently observed in this study and clearly distinguishable. Furthermore, they interact during the polarization reversal and this process has been observed in the PFM images. The phase-field simulation presented in this manuscript further illustrates the interplay between these two types of domains, and the role of the ferroelastic a-domains in the formation of the conductive paths.

7. *"The author mentioned "This results in a 2D electron gas formation responsible for the extraordinary transport properties observed at the 180° -DWs." in the conclusions. The author should explain the two-dimensional electron gas in the corresponding part of the manuscript."*

Answer

The 2D electron gas formation has been explained in the theoretical part of the manuscript. In particular, the simulations suggest a significant band banding associated with a potential drop by 0.8eV at the charged DWs. Assuming the defect level situated around 0.6eV below the bulk conduction band one expects a metallic conduction with the screening described by the Thomas Fermi model, which implies the 2D gas behavior. This description is consistent with pervious publications where the charged DW formation is shown to cause the 2D electron gas formation (Ref. 7).

Reviewer 3

"The authors studied a classical ferroelectric system and found the created 180 domain walls in the a/c domains show giant conduction and designed a device based on this effect. These results are very interesting. I think this manuscript could be published after addressing these issues."

Authors' response: We greatly appreciate the assessment of our work by the reviewer and address below the concerns expressed in the comments.

Comments

1. *"I doubt the metallic-like conduction claimed by the authors. In Fig. 1o, the current is about 90 nA at the bias of 2 V at the room temperature. In Fig. 3k, the current is only about 3 nA at the bias of 4 V at 42 K. Comparing these two values, one could see that the conductivity increase with the increase of temperature. In other words, it behaves like a semiconductor."*

Answer

First, we would like to note that the data collected from the ultra-high vacuum cryogenic AFM-setup are not directly comparable with the conventional Cypher-AFM measurements. It is known from experiments done by different groups that switching and conduction measurements in UHV require a higher voltage e.g. due to a higher barrier at the interface. Additionally, the utilized probes were different, therefore, to allow for an accurate data comparison, it is necessary to compare the temperature dependent measurements within one system using the same probe. For this reason, we conclude that the DW conduction is non thermally-activated, based on our comparison between the cryogenic data at 4K and 42K. In our opinion the 10-fold change of temperature (with little change in conduction) is sufficient for this conclusion and a normal non-degenerated semiconductor is very likely to stop conducting at 4K, which was not the case.

In order to better respond to the concern of the reviewer (and also Reviewer 1) we have added to the Supplementary information (Supplementary Fig. 4) another set of temperature dependent DW conduction measurements with the conventional Cypher-AFM. Here, the temperature was changed between the room temperature and 100°C and the conduction was measured as an averaged value along the DW path. The observed general trend is a decrease of DW conduction vs temperature, which can be partially explained by a polarization decrease and an increase of effective barrier thickness (2D gas retracts further from the surface). All in all, it is very clear that the measured conductivity non thermally-activated as suggested in the manuscript.

Furthermore, we would like to refer to our response to Reviewer 1 where we state that the transport is limited by the potential barrier at the interface between the probe (or electrode) and the PZT. The electron injection from the probe to PZT occurs in the tunneling regime, which is consistent with our experimental observations:

- the measured conduction shows a weak temperature dependence (Fig. 3 from the manuscript and newly measured data discussed below and presented in Supplementary Figure 4)
- the I-V curves are polarity-dependent (Figure 2 in Supplementary information), a significant current is observed only for the positive sample bias

- classic Fowler-Nordheim (FN) formalism provides an adequate description of the measured I-V curve (FN fit is added to the revised Fig. 1 and described in detail in Supplementary Note 5)

However, we acknowledge the validity of the concerns raised by the Reviewer 1 and 3. Because the metallic conduction could not be evaluated directly due to the interface barrier, we decided modify the title replacing the term "metallic" by "non thermally-activated": **"Giant switchable non thermally-activated conduction in 180° domain walls in tetragonal Pb(Zr,Ti)O₃".**

2. *"In the cAFM images at the cryogenic temperatures, there are many bright spots. The authors attribute them to some deposited particles. Why they disappear at the room temperature?"*

Answer

They don't disappear at room temperature but are generally not created in the first place. In the UHV cryogenic setup poling of the sample required a much higher bias compared to the room temperature AFM experiments (8-9V vs. 4V at RT). The particles are deposited on the surface from the tip (or possibly formed due to some electrochemical reactions or mechanical deposition) at voltages >7V, therefore in ambient conditions AFM the problem did not occur.

3. *"The authors say little about the technical details of Fig. 6. Is it a cross section of Fig. 5? Or is it a new simulation (maybe 2D)?"*

Answer

Indeed Figures 5 and 6 represent simulations with two different geometric configurations. At Figure 6 the results of the distinct two dimensional simulation are given. This configuration is a two dimensional slice of PZT thin film with thickness (z-dimension) of 50 nm and width (x-dimension) of 150 nm. All material parameters are the same as for three dimensional case. Such simulation yields exactly the same results as three dimensional modelling (namely, magnitude of polarization, electric potential, elastic strains, bound and free charges), except we have a-domains only in x direction and not in y direction. Such configuration was needed in order to scale down computation and storage requirements to study and visualize the dynamics of domains formation under tip application and after tip release. We have added the description of this configuration to the Methods section.

4. *"Page. 3: "the progressive bending of the domain walls from the charge-neutral head-to-head (HH) or tail-to-tail (TT) 45° orientation". The charge-neutral domain walls should be the head-to-tail type."*

Answer

This is correct, thank you for noticing. We have changed the manuscript accordingly.

5. *"Page. 3: "charged-induced electrostatic potential". It should be "charge"."*

Answer

This has been updated as well. Thanks for noticing.

References

1. Maksymovych, P.; Jesse, S.; Yu, P.; Ramesh, R.; Baddorf, A. P.; Kalinin, S. V. Polarization Control of Electron Tunneling into Ferroelectric Surfaces. *Science* **2009**, *324* (5933), 1421–1425. <https://doi.org/10.1126/Science.1171200>.
2. Nagarajan, V.; Ganpule, C. S.; Li, H.; Salamanca-Riba, L.; Roytburd, A. L.; Williams, E. D.; Ramesh, R. Control of Domain Structure of Epitaxial PbZr_{0.2}Ti_{0.8}O₃ Thin Films Grown on Vicinal (001) SrTiO₃ Substrates. *Appl. Phys. Lett.* **2001**, *79* (17), 2805–2807. <https://doi.org/10.1063/1.1402645>.
3. A.K. Tagantsev, L.E. Cross, J. Fousek, Domains in Ferroic crystals and thin films, ISBN 978-1-4419-1416-3, Springer, 2010
4. Feigl, L.; Yudin, P.; Stolichnov, I.; Sluka, T.; Shapovalov, K.; Mtebwa, M.; Sandu, C. S.; Wei, X. K.; Tagantsev, A. K.; Setter, N. Controlled Stripes of Ultrafine Ferroelectric Domains. *Nat Commun* **2014**, *5*, 4677. <https://doi.org/10.1038/Ncomms5677>.
5. Stolichnov, I.; Feigl, L.; McGilly, L. J.; Sluka, T.; Wei, X. K.; Colla, E.; Crassous, A.; Shapovalov, K.; Yudin, P.; Tagantsev, A. K.; Setter, N. Bent Ferroelectric Domain Walls as Reconfigurable Metallic-Like Channels. *Nano Lett* **2015**, *15* (12), 8049–8055. <https://doi.org/10.1021/acs.nanolett.5b03450>
6. Catalan, G.; Lukyanchuk, I.; Schilling, A.; Gregg, J. M.; Scott, J. F. *J Mater Sci* **2009**, *44* (19), 5307–5311. <https://doi.org/10.1007/s10853-009-3554-0>
7. Sluka, T.; Tagantsev, A. K.; Bednyakov, P.; Setter, N. Free-Electron Gas at Charged Domain Walls in Insulating BaTiO₃. *Nat Commun* **2013**, *4*, 1808. <https://doi.org/10.1038/Ncomms2839>.

REVIEWERS' COMMENTS

Reviewer #1 (Remarks to the Author):

The reviewer is satisfied with the reply and the revision in the manuscript. The reviewer recommends the publication of this manuscript.

Reviewer #2 (Remarks to the Author):

The authors responded well to the reviewers' questions, I recommend this manuscript for publication in Nature communications.

Reviewer #3 (Remarks to the Author):

I am satisfied with this version.